# Prevalence of Comorbidities and Associated Factors among HIV Patients Attending Antiretroviral Clinics in the Tamale Metropolis, Ghana

**Kingsley Aseye Hattoh [1],\*** , **Bryan Aapentuo Sienso [2]** and **Eugene Dogkotenge Kuugbee [3]**

1 Department of Paediatrics, Tamale Teaching Hospital, Tamale P.O. Box TL 16, Ghana
2 Department of Public Health, Tamale Teaching Hospital, Tamale P.O. Box TL 16, Ghana
3 School of Medicine and Dentistry, CK Tedam University of Technology and Applied Sciences, Navrongo P.O. Box 24, Ghana
\* Correspondence: hattohkingsley@gmail.com; Tel.: +233-547986557

**Abstract:** Human Immunodeficiency Virus (HIV) is a chronic infectious disease, and without antiretroviral therapy (ART), it is associated with comorbidities. The prevalence of comorbidities, adherence to ART and quality of life (QoL) of HIV patients were studied. A cross-sectional study design involving 360 HIV patients from two ART clinics across the Tamale metropolis was employed. Socio-demography, adherence to therapy, and QoL data were taken with the help of a questionnaire and analyzed using SPSS version 24. The prevalence of comorbidities was 30.3% (109), with Hepatitis B infection (73, 20.3%) being the most prevalent. Adherence levels were high (192, 53.3%), moderate (108, 30.0%) and low (60, 16.7%). Overall, QoL was excellent amongst 149 (41.4%), good in 169 (46.7%), and poor in 42 (11.7%) respondents. Marital status, presence or absence of HIV symptoms, adherence level to ART and overall QoL, especially Physical, Psychological, and independence domains ($p < 0.05$), were factors associated with the presence of comorbidities. There is a high level of comorbidities among persons living with HIV (PLWH) in the Tamale metropolis influenced by QoL and adherence to ART. We recommend a multifaceted approach to the management of PLWH.

**Keywords:** human immunodeficiency virus; comorbidities; quality of life

## 1. Introduction

The life expectancy of persons living with human immunodeficiency virus (HIV) and or acquired immunodeficiency syndrome (AIDS) has recorded an upward trend over the past few years. This has largely been credited to the implementation of highly active antiretroviral therapy (HAART) amongst these persons living with human immunodeficiency virus (PLWH) [1]. Combined antiretroviral therapy (cART) usage has made HIV a chronic infectious disease. Thus, persons living with HIV are expected to have a normal life expectancy, as depicted by some survival models [2]. There is, however, an emerging trend of comorbidities among PLWH compared to those without HIV [3]. The question that has lingered for a while now is whether the upward trend of life expectancy among PLWH has any influence on these persons developing comorbidities like liver and heart disease, mental health issues, cancer and neurocognitive impairments for which there is no direct link to HIV [4]. Low CD4 counts have been found to predispose PLWH to comorbidities. This has been attributed to non-adherence to antiretroviral therapy (ART) [5]. These persons have been found to have a lower quality of life (QoL) which predisposes to and may worsen a comorbid state [6]. A good understanding of comorbidities will help improve the long-term management of PLWH [7]. This research work brings to light the prevalence of comorbidities among PLWH and how adherence to ART influences this. Also, the QoL of PLWH was explored to ascertain what can be incorporated into the current HIV/AIDS program to improve the QoL of these persons. This study, though conducted in the Tamale

Metropolis, can serve as a blueprint for similar studies across the country. This will help make meaningful changes to current HIV or AIDS programs, to incorporate management protocols, for the problems that come with increasing life expectancy.

## 2. Materials and Methods

The study was conducted in the Tamale Metropolis, a district in the Northern Region of Ghana. Tamale Teaching Hospital and Tamale Central Hospital are the two hospitals in the Metropolis whose ART clinics were used as study sites. Routine clinic visits for these two centers range from one to two times a month for each person for review, counseling and refill of medications. A quantitative approach was used for the study. The study design was observational with an analytic study. A cross-section of PLWH in the Tamale Metropolis was studied to examine the prevalence of comorbidity, their QoL and adherence to ART. The study was conducted from July 2021 to September 2021. Samples were from any person living with HIV above the age of 18 who came to the ART clinics. Yamane's formula (n= $\frac{N}{1+Ne2}$) was used to calculate a sample size of 360 participants who were recruited from the Tamale Central Hospital and Tamale Teaching Hospital for inclusion in the study. From the formula, n = minimum sample size, N = total population (population living with HIV in the northern region as of 2019) and e = the margin error (0.05) because the formula was set for a minimum sample size at 95% confidence level [8,9]. Purposive sampling was used to select those who had tested positive for HIV, were on ART and visited the ART clinic for routine care and follow-up [10].

Data were collected from participants mainly via a google form on electronic gadgets such as phones or tablets. The questionnaire was developed by adapting standard questionnaires [11]. The questionnaire was made up of socio-demography, the 8-Item Morisky medication adherence scale (MMAS-8) and the world health organization's quality of life for HIV BREF (WHOQOL-HIV BREF) tool (Supplementary Materials).

The MMAS-8 scale is a self-reported medication adherence measure that was developed from an earlier approved 4-item scale. The questions are phrased in such a way that questions 6 and 7 are rephrased to avoid the "yes-saying" bias. The first seven questions require a yes or no answer. However, the eighth question assumes a 5-point Likert response [12]. Though each question is not a determinant of adherence, it measures specific medication-taking behavior [13].

The WHOQOL-HIV BREF assessment tool assessed the QoL of participants. This tool is the abridged form of the original version of WHOQOL-HIV, which constitutes 120 questions and Likert-scale responses [14]. The responses to these questions described the perception of an individual's position in life with regard to their culture and value systems in respect of their goals, expectations, standards and concerns [15]. The WHOQOL-HIV BREF is made up of 31 questions couched from and representative of the 120 questions. These questions were grouped into six domains. These are the physical, psychological, level of independence, social relationship, environment and spiritual or religious or personal beliefs domain [16].

Comorbidities were sampled from a literature review of similar studies and adopted in our studies. This was defined as any disease which is not a part of the AIDS spectrum in a PLWH [17]. An option was provided for other comorbidities not listed specifically in our questionnaires. Comorbidities were confirmed by respondents' laboratory results, where applicable, with most of them showing us medications they were on for the treatment of these comorbidities. The number of years one was living with HIV was determined based on when respondents could recall they first had the infection. Most of the respondents showed test results that showed the dates they first tested positive for HIV. The decision to separate them into those who have known or been living with the infection for less or more than five years was informed by a careful literature review on the subject [18]. Symptoms such as prolonged fever, unexplained fatigue, swollen lymph nodes, rapid weight loss and profuse night sweats reported by respondents helped in classifying them into symptomatic or asymptomatic [19].

The questionnaires were structured such that there could be self-reporting. However, there were trained personnel who clarified questions respondents had as well as interviewed and filled in the responses of most study participants. The questionnaires were entered on a Google form and exported to an Excel spreadsheet. The data was cleaned on the Excel spreadsheet by vetting responses and discarding incomplete responses. Responses were coded following standard protocols and exported to SPSS version 24.

The responses on the Likert scale for the WHOQOL-BREF questionnaire were coded as 1 for very poor, 2 for poor, 3 for neither poor nor good, 4 for good and 5 for very good quality of life [20,21]. However, the negatively-phrased questions in both the WHOQOL-BREF and the MMAS-8 were reversed and recoded. The domain scores were calculated by averaging the facet scores and multiplying them by 4 to make them comparable to the full version of the WHOQOL tool [22]. The QoL scores were categorized into excellent, good and poor. The percentages of each score were found, which were calculated by dividing a transformed score ($\leq$20) by the ideal total of a transformed score (20) for a domain and multiplying by 100%. Any score percentage from 80% to 100% was classified as an excellent QoL. A score percentage from 60% to 79% qualified one as having a good QoL, and a score below 60% was a poor QoL [23]. The various domains were then cross-tabulated against the various variables, and the frequencies were found and tabulated [24].

The MMAS-8 was categorized into low, moderate and high levels of adherence [25]. The scores were tallied from the responses received. A score of 1 was apportioned to a 'no' and 0 to a 'yes'. The eighth question had a 5-point Likert scale response. 'Never' represented 4 points, 'once in a while' 3 points, 'sometimes' 2 points, 'usually' 1 point, and 'all the time' 0 points. A total of 11 to 9 points was expected for a high level of adherence. The moderate level had scores from 8 to 5 points. A cumulative score of 4 and below characterized a low level of adherence [26,27].

A Chi-square test with Fisher's exact correction was used during the analysis. Results were presented in tables with a cross-tabulation of explanatory variables.

Ethical clearance was given by the Ghana Health Service Ethics Committee with approval number GHS-ERC 028/07/21 and the Committee on Human Research, Publication and Ethics with approval number CHRPE/AP/202/21. Consent was sought from respondents as well. The questionnaire was devoid of names and serial numbers that could link information to specific individuals. This improved the confidentiality of the study. Only the Investigators could access the raw data gathered from the study.

## 3. Results

The description of the experimental results, their interpretation, as well as the experimental conclusions that were drawn, are seen in the subsections.

### 3.1. Socio-Demographic Characteristics

The results, as represented in Table 1, showed that the majority of the people who attended the ART clinics were females (77.5%) and between the ages of 40 to 49 years (31.4%), with a median age of 40 years. The mean age of these respondents was 41.46 $\pm$ 11.3 years. The majority of respondents were married (52.8%), with those divorced (6.4 %) constituting the least percentage in the group. The majority of the respondents had no formal education (33.3%), trailed by 28.9% studying up to the tertiary level, then secondary (26.9%) and primary education (10.8%), respectively.

**Table 1.** Socio-Demographic Characteristics.

| Variable | Frequency (N = 360) | %N |
|---|---|---|
| **Sex** | | |
| Male | 81 | 22.5 |
| Female | 279 | 77.5 |
| **Age groups** | | |
| 18–29 | 53 | 14.7 |
| 30–39 | 108 | 30.0 |
| 40–49 | 113 | 31.4 |
| >49 | 86 | 23.9 |
| **Marital Status** | | |
| Married | 190 | 52.8 |
| Divorced | 23 | 6.4 |
| Widowed | 66 | 18.3 |
| Single | 81 | 22.5 |
| **Educational Status** | | |
| None | 120 | 33.3 |
| Primary | 39 | 10.8 |
| Secondary | 97 | 26.9 |
| Tertiary | 104 | 28.9 |
| **Presence/absence of HIV symptoms** | | |
| Asymptomatic | 337 | 93.6 |
| Symptomatic | 23 | 6.4 |
| **Duration of HIV Infection (since time of infection)** | | |
| <5 years | 195 | 54.2 |
| >5 years | 165 | 45.8 |

A total of 23 respondents were symptomatic for HIV, with a lot more males (12.3%) being symptomatic for HIV compared to females (4.7%). About 54.2% of the respondents have been living with the infection for less than five years. However, 45.8% had been living with the disease for more than five years since the time of infection.

### 3.2. Prevalence of Comorbidities

The number of PLWH, together with other comorbidities, was 109 (30.3%) as presented in Table 2. The majority, 101 (28.1%), of those who had comorbidities had just one comorbidity, with the remaining few having 2 or 3 comorbidities, 8 (2.2%), in addition to HIV. Out of the 360 respondents, 73 (20.3%) respondents were living with Hepatitis B infection. 33 (9.2%) were hypertensive, 11(3.1%) had peptic ulcer disease (PUD), 4 (1.1%) had other forms of liver pathology, including Hepatitis A, C and fatty liver disease, 4 (1.1%) had dyslipidemia, and 8 (2.2%) had other forms of cardiovascular diseases(CVD) including acute coronary syndrome (ACS), previous myocardial infarct and cardiomyopathy, 4 (1.1%) had mental health issues, 3 (0.8%) were diabetic, and 2 (0.6%) were living with asthma. None of the respondents suffered a stroke.

### 3.3. Adherence to Antiretroviral Therapy

The level of adherence of respondents to ART is classified as low, moderate and high. These were based on scores from the MMAS-8. The majority of respondents, numbering 192 (53.3%) patients, were reported as being highly adherent to their medications. Moderate and low adherence levels were recorded as 108 (30.0%) and 60 (16.7%) respondents, respectively, as shown in Figure 1.

**Table 2.** Prevalence of Comorbidities.

| Variable | Frequency (N) | %N |
| --- | --- | --- |
| Comorbidity present | 109 | 30.3 |
| Comorbidity absent | 188 | 69.7 |
| Total | 297 | 100 |
| One comorbidity present | 101 | 28.1 |
| Multiple comorbidity | 8 | 2.2 |
| Total | 109 | 30.3 |
| **Comorbidities** | | |
| Hypertension | 33 | 9.2 |
| Hepatitis B | 73 | 20.3 |
| Other liver disease (fatty liver, HCV) | 4 | 1.1 |
| Peptic ulcer | 11 | 3.1 |
| Cardiovascular diseases | 8 | 2.2 |
| Dyslipidemia | 4 | 1.1 |
| Diabetes | 3 | 0.8 |
| Mental health issues | 2 | 0.6 |
| Asthma | 2 | 0.6 |

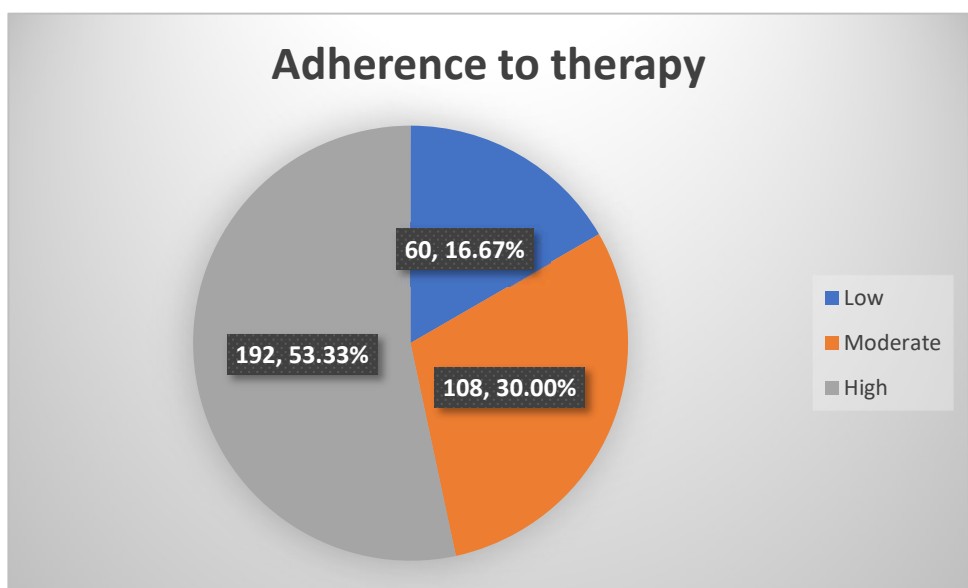

**Figure 1.** Adherence to anti-retroviral therapy.

*3.4. Quality of Life*

The QoL of PLWH in the Tamale Metropolis is presented in Table 3. The psychological domain had the highest number of participants scoring an excellent QoL. The psychological domain assessed the bodily image and appearance of respondents, presence of negative and positive feelings, self-esteem, thinking, learning, memory and concentration of the respondents, which gave an excellent cumulative score for 309 (91.2%) of the respondents. The scores in this domain averaged 17.8 ± 1.9, with a minimum score of 12 and a maximum score of 20. The domain representing the environment was seen as having the least number of respondents scoring excellent for QoL. The environmental domain includes financial resources, healthcare availability, opportunities for acquiring new information and skills, opportunities for leisure, and transport. The number of participants who had excellent domains for the environment was 106 (31.5%). However, a greater number of respondents, constituting 229 (68.2%), scored 'good' for QoL, and 1 person (0.3%) scored poor for the environment domain. The average score in the environment domain is 14.6 ± 1.9, with 9 being the minimum score and 18.5 being the maximum.

**Table 3.** Quality of Life of participants.

| | | Quality of Life | | |
|---|---|---|---|---|
| **Domain** | **Poor** | **Good** | **Excellent** | **Average (min–max)** |
| Physical | 4 (4.4) | 41 (11.4) | 315 (87.5) | 18.1 ± 2.1 (8–20) |
| Psychological | 0 | 30 (8.8) | 309 (91.2) | 17.8 ± 1.9 (12–20) |
| Independence | 0 | 72 (20.0) | 288 (80.0) | 17.4 ± 2.3 (11–20) |
| Relationship | 3 (0.8) | 106 (29.4) | 251 (69.7) | 16.7 ± 2.4 (8–20) |
| Environment | 1 (0.3) | 229 (68.2) | 106 (31.5) | 14.6 ± 1.9 (9–18.5) |
| Spirituality | 24 (6.7) | 118 (32.8) | 218 (60.6) | 15.8 ± 3.7 (6–20) |
| Overall Health | 27 (7.5) | 180 (50.0) | 153 (42.5) | 4.3 ± 0.7 (1–5) |
| Overall QoL | 42 (11.7) | 169 (46.9) | 149 (41.4) | 4.3 ± 0.7 (1–5) |

The overall QoL was excellent for 149 (41.4%), good for 169 (46.9%) and poor for 42 (11.7%) of the respondents. The overall QoL assessed how well respondents viewed their QoL. The overall QoL averaged 4.3 ± 0.7 with a minimum score of 1 and a maximum score of 5. The general health of respondents was excellent for 153 (42.55), good for 180 (50%), and poor for 27 (7.5%) of the respondents. Their average score is 4.3 ± 0.7, with a minimum score of 1 and a maximum score of 5.

### 3.5. Factors Associated with the Occurrence of Comorbidities among HIV Patients

There were a number of factors studied for association with the presence of comorbidities amongst PLWH, as represented in Table 4. These factors include socio-demographic status (age, education, gender, marital status, presence or absence of HIV symptoms, and the number of years one had been living with HIV), adherence to ART and the WHO-QOL of the respondents. The statistical significance of these associations is also indicated in Table 4.

**Table 4.** Presence of comorbidities and associated factors.

| | Presence of Comorbidities | | | |
|---|---|---|---|---|
| **Variable** | **Yes (%)** | **No (%)** | $\chi^2$ | ***p*-Value** |
| **Age** | | | 2.1 | 0.54 |
| 18–29 | 12 (3.3) | 41 (11.4) | | |
| 30–39 | 32 (8.9) | 76 (21.1) | | |
| 40–49 | 38 (10.6) | 75 (20.8) | | |
| ≥50 | 27 (7.5) | 59 (16.4) | | |
| **Education** | | | 1.8 | 0.61 |
| None | 36 (10.0) | 84 (23.3) | | |
| Primary | 14 (3.9) | 25 (6.9) | | |
| Secondary | 25 (6.9) | 72 (20.0) | | |
| Tertiary | 34 (9.4) | 70 (19.4) | | |
| **Gender** | | | 3.2 | 0.07 |
| Male | 18 (5.0) | 63 (17.5) | | |
| Female | 91 (25.3) | 188 (52.2) | | |
| **Marital Status** | | | 12.6 | 0.005 |
| Married | 54 (15.0) | 136 (37.8) | | |
| Divorced | 9 (2.5) | 14 (3.9) | | |
| Widowed | 30 (8.3) | 36 (10.0) | | |
| Single | 16 (4.4) | 65 (18.1) | | |
| **Presence or absence of symptoms of HIV** | | | 22.2 | <0.001 |
| Asymptomatic | 92 (25.6) | 245 (68.1) | | |
| Symptomatic | 17 (4.7) | 6 (1.7) | | |

**Table 4.** *Cont.*

| | Presence of Comorbidities | | | |
|---|---|---|---|---|
| **Variable** | **Yes (%)** | **No (%)** | $\chi^2$ | ***p*-Value** |
| **Years living with HIV (since time of infection)** | | | 0.20 | 0.73 |
| ≤5 years | 61 (16.9) | 134 (37.2) | | |
| >5 years | 48 (13.3) | 117 (32.5) | | |
| **Adherence to therapy** | | | 9.45 | 0.009 |
| Low | 27 (7.5) | 33 (9.2) | | |
| Moderate | 35 (9.7) | 73 (20.3) | | |
| High | 47 (13.1) | 145 (40.3) | | |
| **QoL Physical domain** | | | 12.46 | 0.002 |
| Poor | 4 (1.1) | 0 | | |
| Good | 17 (4.7) | 24 (6.7) | | |
| Excellent | 88 (24.4) | 227 (63.1) | | |
| **QoL Psychological domain** | | | 10.16 | 0.002 |
| Good | 17 (5.0) | 13 (3.8) | | |
| Excellent | 88 (26.0) | 221 (65.2) | | |
| **QoL Independence domain** | | | 6.96 | 0.010 |
| Good | 31 (8.6) | 41 (11.4) | | |
| Excellent | 78 (21.7) | 210 (58.3) | | |
| **QoL Relationship domain** | | | 0.61 | 0.76 |
| Poor | 1 (0.3) | 2 (0.6) | | |
| Good | 29 (8.1) | 77 (21.4) | | |
| Excellent | 79 (21.9) | 172 (47.8) | | |
| **QoL Environment domain** | | | 0.47 | 0.79 |
| Poor | 0 | 1 (0.3) | | |
| Good | 69 (20.5) | 160 (47.6) | | |
| Excellent | 33 (9.8) | 73 (21.7) | | |
| **QoL Spirituality Domain** | | | 4.48 | 0.11 |
| Poor | 9 (2.5) | 15 (4.2) | | |
| Good | 43 (11.9) | 75 (20.8) | | |
| Excellent | 57 (15.6) | 161 (44.7) | | |
| **General Health status** | | | 7.13 | 0.028 |
| Poor | 9 (2.5) | 18 (5.0) | | |
| Good | 43 (11.9) | 137 (38.1) | | |
| Excellent | 57 (15.8) | 96 (26.7) | | |
| **Overall QoL** | | | 6.60 | 0.038 |
| Poor | 15 (4.2) | 27 (7.5) | | |
| Good | 40 (11.1) | 129 (35.8) | | |
| Excellent | 54 (15.0) | 95 (26.4) | | |

Socio-demographic factors influencing the occurrence of comorbidities were marital status and the presence or absence of HIV symptoms ($p < 0.05$). However, age, education, gender and the number of years one lived with HIV had no association with the presence of comorbidities ($p > 0.05$), as indicated in Table 4.

Adherence to ART was seen to have influenced the presence of comorbidities ($p < 0.05$).

The general health status of patients and overall QoL influenced the occurrence of comorbidities among HIV patients ($p < 0.05$). Among the domains of QoL, physical, psychological, and independence domains were found to influence the occurrence of comorbidities in HIV patients ($p < 0.05$). Thus, how comfortable one was, their energy and rest levels, positive and negative feelings, bodily image and self-esteem were all seen to have an impact on the presence of comorbidities.

## 4. Discussion

### 4.1. Prevalence of Comorbidities

In recent times, PLWH have been noted to have increased life expectancy based on various studies [28,29]. However, with increasing life expectancy has come an increasing prevalence of comorbidities, as seen in our study. The prevalence of comorbidities among PLWH from our study is 30.3%, with the majority (28.1%) having just one comorbidity and the remaining few (2.2%) having two or three comorbidities. This is consistent with findings from [7,17,30,31]. The presence of comorbidities could be associated with the fact that with increasing life expectancy, HIV/AIDS has evolved into a chronic infectious disease.

Hepatitis B, reported by 73 (20.3%) respondents, was the most prevalent comorbidity in our study. A similar study carried out in Brent, North West London, also found Hepatitis B to be the most prevalent comorbidity in 69 respondents [32]. This may be due to the immune deficiency due to HIV/AIDS and, in some cases, due to the similar use of intravenous drugs as a risk factor, which was not studied.

Hypertension for 33 respondents (9.2%) was the next prevalent comorbidity. This finding is similar to studies carried out across various communities in Ghana and Manaus, Brazil, where about 36% of the respondents had hypertension [33,34]. This is about four times what we recorded in our study. This could be because our study participants did not routinely check their blood pressure levels as compared to the participants in the other studies who were actively checked and diagnosed. In the other studies, the prevalence of hypertension was linked to cART.

PUD was reported among 11 respondents, which is consistent with findings from a study in Kumasi, Ghana [35]. However, most studies attribute PUD to H. pylori infection rather than HIV/AIDS, though there are a few that believe the presence of HIV/AIDS makes the host more susceptible to the H. pylori infection. Anorexia, which is a side-effect of some ART, doubles as a predictor for PUD [36,37].

Dyslipidemia was reported amongst four (1.1%) respondents. This contradicts other studies that found a very high number of study participants having dyslipidemia. A study carried out in North Shewa, Ethiopia, found a high prevalence of dyslipidemia (59.9%) among HIV-infected patients on first-line ART [38]. Our figures may have been lower because patients did not routinely check their lipid profiles. There may be a lot of asymptomatic patients for dyslipidemia amongst our study participants but may not know about it. However, there was another study done in Western Kenya that found no association between dyslipidemia and HIV though figures for dyslipidemia were high as well (47%) [39].

CVDs, which included acute coronary syndrome, cardiomyopathies and previous myocardial infarction, were reported among eight (2.2%) respondents. This is about twice the number for dyslipidemia. This is not surprising because dyslipidemia is one of the commonest and known risk factors for CVDs. Hypertension is also a known risk factor that was reported by most of our respondents. The CVDs reported could be due to the above risk factors, as seen in other works such as [40–43], to mention but a few.

Diabetes was reported by 3 (0.8%) of our study participants. This is consistent with other studies, in it being present, but contradicts them in diabetes being the most prevalent in a lot of these studies. Most of these studies recorded larger percentages, around 8 to 14%, compared to 0.8% in our studies. Most of the participants in these other studies developed insulin-resistance diabetes mellitus and were all noted to be on ART [44–48]. We cannot say the same for our study because we did not include a separate study of the association between diabetes and ART.

Mental health issues, reported in 2 (0.6%) respondents, were seen in other studies. The statistic from our study ranged between 2% less than what was reported by [49] and 70% less than another done in the Central Region of Ghana by [50]. This could be due to the culture of silence on mental health issues in our part of the world. Also, some of the respondents may not have had insight into their mental health conditions since mental

health is a delicate issue to broach and is not easily identifiable except if there are overt signs or it is diagnosed by a trained psychiatrist. In other studies, HIV was found to increase the severity of mental health issues [51].

There were other forms of liver diseases, including A, C and fatty liver disease, reported by 4 (1.1%) of our respondents. This is consistent with other studies where various forms of liver diseases are found in HIV-infected patients, especially coinfections with Hep B or C [52–54]. This is expected because even amongst the general population in the Tamale Metropolis, liver diseases have been noted to be on the rise though HIV is being linked to increasing severity [55,56]. This is even worse because some of the ARTs are known to be hepatotoxic, thus creating the problem of the health service provider getting to their wit's end on deciding the choice of drugs, especially in cases of deranged liver function test results.

Asthma was reported by two of our respondents. This number is much smaller compared to a study done in Uganda, where the numbers are about 20% more than what we recorded [57]. In a similar study, asthma was found to be the most prevalent non-infectious pulmonary comorbidity among PLWH [58].

### 4.2. Adherence to Antiretroviral Therapy

The majority of the respondents were found to be adherent to their ART based on the MMAS-8 they recorded. About 53.3%, 30% and 16.7% scored high, moderate and low adherence, respectively. In a similar study, 59 respondents or 23.4%, had low adherence, consistent with the findings in this study [59]. However, findings from a few other studies contradicted ours [60,61]. These other studies had as high as 40% of their respondents not adhering to treatment. Our study may have recorded high levels of adherence because these are respondents who attend the antiretroviral clinic and the directly observed therapy instituted to improve patient adherence to medications. This in itself may influence their adherence to treatment.

### 4.3. Quality of Life

In our study, a lot of the respondents, 309 (91.2%), had excellent scores for QoL in the psychological domain compared to the other domains, consistent with findings of [62], who also studied PLWH across HIV centers of excellence in India. However, studies by [63–65] in Akure, situated in Nigeria, and Ho, situated in Ghana and Ethiopia, respectively, contrasted ours. Our findings may have recorded the highest averages for the psychological domain due to the presence of an effective counselling unit and models of hope. Models of Hope is a non-governmental organization that has recruited some PLWHs who serve as models for patients who visit the ART clinic. They facilitate access to healthcare while providing psychological and emotional support to PLWH.

The environmental domain had the least number of respondents scoring excellent and also the least average score. The majority, however, had good scores. This is consistent with findings by [66,67]. However, the majority of the respondents attest to having access to healthcare which constitutes the environment domain. Most of these persons had issues with finances, opportunities for acquiring new information and skills, and opportunities for leisure. These impacted negatively on the average score. All these may also be due to the low-to-middle-income status of the country.

The majority of the respondents had good QoL, followed by excellent QoL, with the least recording poor QoL. This is similar to the findings of a study by [68] done in Togo. This may be due to the perception of the respondents that once there is a harmful organism living in their bodies, their system may not be as it used to be without the organism.

### 4.4. Prevalence of Comorbidities and Associated Factors

The factors that may be associated with the prevalence of comorbidities from our study include marital status, presence or absence of HIV symptoms, adherence to antiretroviral therapy and some domains under the QoL. These domains are the physical, psychological,

independence, overall QoL and general health status. The majority of the factors in our study are consistent with studies by [7,17,69] which all saw an association between ART, the presence or absence of HIV symptoms and the prevalence of comorbidities. Our findings did not have a significant association between the respondents' ages or the number of years the patient has been living with HIV since the time of infection and the prevalence of comorbidity, unlike what is seen in studies by [3,70–72].

Marital status findings in association with the prevalence of comorbidities in our study are consistent with findings by [73]. That study had the married being at lesser risk of comorbidity compared to the widowed and the divorced but did not entirely agree with our study in that the monogamous married respondents showed a lesser prevalence of comorbidities compared to those who were never married or single. Our study, however, did not find out if those married were in monogamous or polygamous marriages. This could have influenced our finding hence the contradiction. Those who were widowed may be seen being more prevalent with other comorbidities because most of them may get to know their status at the passing of their spouse, by which time their immunity is low. Thus, exposing them to comorbidities and other infectious diseases that are in the pathway of some of these comorbidities. Also, the psychological effect of the loss of a loved one could also have deleterious health effects that may contribute to some of these comorbidities.

The effects of having AIDS or symptoms of AIDS are seen with an increasing prevalence of comorbidities. This is consistent with a lot of studies all over the world [74–76]. This is attributed mainly to the immune-mediated inflammatory processes that are associated with the infection, thus predisposing PLWH to comorbidities. Our study, however, could not tell if the comorbidities were there prior to the HIV infection or after. This would have helped us know whether the infection predisposed to comorbidities or the presence of comorbidities may rather impact their health.

Our study shows that adherence to antiretroviral medications reduces the prevalence of comorbidities. Thus suggesting that adherence to antiretroviral medications may be protective against comorbidities. In a similar study by [77], antiretroviral was not only seen reducing the prevalence of comorbidities but may have been responsible for slowing the progress of some comorbidities such as Hep B and Hep C. This is attributed to the fact that the causative agents of some of these comorbidities are viral, thus the efficacy of the medication in those conditions as well. Another study by [78] showed a reduction in the prevalence of comorbidities in those on antiretroviral medication, while those on traditional medicine had a two-fold increase in comorbidities. A few other studies contradicted our study. They did not find an association between antiretroviral medications and the prevalence of comorbidities [70]. Some also attributed the presence of comorbidities to the use of antiretroviral medications with an emphasis on the line of the drug, drug-drug interactions and duration of treatment [3,79,80].

The physical, psychological, and independent domains, as well as the overall QoL and general health status, had a statistically significant impact on the presence of comorbidities. The lower the QoL, the higher the prevalence of comorbidities. This is consistent with studies done by [6,74,81,82]. This goes to explain how one's day-to-day activities and the use or overuse of substances can impact these comorbidities. For instance, one of the most common risk factors of hypertension is lifestyle which is influenced by sleep and rest levels [83]. There is a link between treatment efficacy and psychological factors, according to other studies [84,85]. This supports our findings of an excellent psychological domain translating into a decreased prevalence of comorbidities. This can also be attributed to the incorporation of a counseling unit in the ART clinic set-up as well as the presence of Models of Hope. Again, this goes to explain the high perception rates of the overall QoL and the general health of the respondents.

## 5. Conclusions

- This study confirmed and is consistent with our belief that comorbidities are prevalent in PLWH. This is in line with the research objective. Although from the literature,

comorbidities were expected to be prevalent in this group of persons, 30.3% was rather thought to be on the high side for persons who were not actively (that is, taking samples or conducting clinical examinations or measurements on the spot) and routinely being checked for the presence of comorbidities at the ART clinic. This means comorbidities are prevalent amongst PLWH in the Tamale Metropolis. Thus, with active and routine checking, the numbers may be increased, and give individuals the opportunity to seek and get treatment early. Hepatitis B as the most prevalent comorbidity did not come as a surprise because that is what has been widely studied and seen.

- The adherence to ART being high in more than half of the population is in line with the research objective and expected outcomes. This is because these are people who routinely visit the ART clinic and are expected to be concerned about their health and well-being. For such persons, we expect them to be adherent to their medications, and the QoL was excellent for the majority of the respondents in the physical, psychological, independence, relationship, and spiritual domains. The overall QoL and health are good, which translates to an average or moderate score for the majority of the respondents. This is expected because, with adherence and routine check-ups, we assume the QoL of persons should be good, if not excellent.

- The presence of comorbidities seen may have been influenced by marital status, the presence or absence of HIV symptoms, adherence to antiretroviral medications, and the quality of life in the physical, psychological and independence domains. The overall QoL and general health status was one of the justifications for this study. That is, to look at the interplay between these factors. These fulfilled our research objectives and are findings that are consistent with our thoughts at the onset of this study. This can help persons living with the disease to adopt lifestyles healthy enough to help them live close to normal healthy, quality lives. The best form of treatment is prevention. These findings will help us champion the course of preventing comorbidities in these group of persons now that our findings allude to it.

**Supplementary Materials:** The following supporting information can be downloaded at: https://www.mdpi.com/article/10.3390/venereology2010001/s1, more details information can be downloaded at: www.moriskyscale.com, https://apps.who.int/iris/handle/10665/77775.

**Author Contributions:** Conceptualization, K.A.H. and E.D.K.; Data curation, B.A.S.; Formal analysis, E.D.K.; Investigation, K.A.H. and B.A.S.; Methodology, K.A.H.; Project administration, K.A.H. and B.A.S.; Resources, K.A.H.; Software, E.D.K.; Supervision, B.A.S. and E.D.K.; Validation, B.A.S. and E.D.K.; Writing—original draft, K.A.H.; Writing—review and editing, B.A.S. and E.D.K. All authors have read and agreed to the published version of the manuscript.

**Funding:** This research received no external funding.

**Institutional Review Board Statement:** The Committee on human research, publication and ethics considered the ethical merit of our submission and approved the protocol. The approval number is CHRPE/AP/202/21. The Ghana Health Service Ethics Review Committee has reviewed and given approval for the implementation of our study protocol. The approval number is GHS-ERC 028/07/21. The study was conducted in accordance with the Declaration of Helsinki and approved by the Ethics Committees of the Committee on Human Research, Publication and Ethics (CHRPE/AP/202/21 and 15 June 2021) and the Ghana Health Service Ethics Review Committee (GHS-ERC 028/07/21 and 30 August 2021).

**Informed Consent Statement:** Informed consent was obtained from all subjects involved in the study. There is written informed consent as well for the publication of this work.

**Data Availability Statement:** Not applicable.

**Acknowledgments:** We acknowledge the conducive environment created by the staff of both antiretroviral clinics of Tamale Teaching Hospital and Tamale Central Hospital.

**Conflicts of Interest:** The authors declare no conflict of interest.

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
