# Peer review of "Prevalence of Comorbidities and Associated Factors among HIV Patients Attending Antiretroviral Clinics in the Tamale Metropolis, Ghana"

_venereology, doi:10.3390/venereology2010001_

Round 1

Reviewer 1 Report

INTRODUCTION: 

1 - A careful review of acronyms is required. For example: Line 34: "persons living with human immunodeficiency virus (PLHV)". In line 35 there, for example, is the same expression "people living with human immunodeficiency virus" instead of the acronyms (PLWHV).

2 - I suggest the expression "chronic infectious disease" (line35) instead of just "chronic disease".

3 - In the introduction, the question from line 43 to line 54 (up to reference 9) is an important topic, but it is out of context with the other questions in the paragraph. I suggest reviewing how to place this information in the introduction for better understanding or even deleting it due to the common issue of introduction length.

4 - As the occurrence of comorbidities is associated with adherence to ART and quality of life, I suggest more information about these questions in the introduction topic, instead of questions from line 43 to line 54, for example.

5 - The sentence from line 56 to line 58 is unnecessary because it has the same information as the sentence in lines from 34 to 37

REFERENCES:

1 - There is no references or even a brief explanation on Yamane, Morisk and WHOQOL-HIV BREF tools

MATERIALS AND METHODS AND RESULTS

1 - There is no value of confidence interval

2 - There is no description or definition of some exposure or outcome variables. For example: What is the difference between asymptomatic and symptomatic serostatus?

3 - Defining comorbidity is challenging for considering the lack of consensus about how to identify, measure and how patients with multiple conditions view their illness. Thus, I think that considering the comorbidity occurrence only through an interview without any exam or medical chart or even medicatin prescription may affect the results. This issue represents a great limitation in the study and must be mentioned.

DISCUSSION:

1 - I consider hepatitis a co-infection and not a comorbidity. As mentioned in lines 209 to 212, both infections carry the same risk of contagion.

2 -  I think the study brings a lot of different information and, therefore, it is not possible to discuss them in depth. I suggest choosing between adherence to ART or quality of life.

3 - As mentioned in the introduction section, people living whit HIV on ART as long as live they may have comorbidities. Thus, an important fact that was not discussed  was  the variable "years living whit HIV". The study did not show the median or mean for this variable. As the majority of individuals presented  < 5 years, I concluded that the respective median or mean could be low.  This fact could explain the lower comorbidity incidence and must be discussed.

Author Response

  1. A careful review of acronyms has been done to include relevant and appropriate acronyms throughout the text where necessary. 
  2. The expression chronic infectious disease has been used instead of chronic disease as suggested 
  3. The question placed in the previous version of the manuscript from line 43 to 54 has been deleted as suggested.
  4. In place of the question from line 43 to 54 is some additional information on the occurrence of comorbidities and its association to adherence to antiretroviral therapy and quality of life.
  5. The sentence from line 56 to 58 was deleted after it was pointed out by referee 1 to contain same information as line 34 to 37.
  6. The Yamane formula, 8-point Morisky’s Medication Adherence Scale and WHOQOL-HIV BREF tools have been duly referenced and explained under the materials and methods section as recommended.
  7. The value of confidence interval of 95% used in the analysis has been duly stated in the manuscript as recommended.
  8. Comorbidity has been defined under the materials and methods section. Information on how data on comorbidity was reported was documented as well as requested. Limitations recognized when compared to other works have been stated during discussion.
  9. Based on the definition of comorbidity and review of various literature on the subject, the co-infection of Hepatitis B with HIV is considered a comorbidity and has been indicated in the revised manuscript.
  10. Regarding ART and QoL, though they could be described separately, ART could affect QoL and presence of comorbidities as we hypothesized and is reflected in our results as well. Hence, we described how the ART, QoL and comorbidities are related to drive the importance of the multifaceted approach to improving the health of PLWH as we recommended.
  11. The variable, years living with HIV since infection, was found not having statistical significance from the p-value recorded during analysis thus, its discussion was not detailed.
  12. Meaning of participant being symptomatic and asymptomatic for HIV has been clearly described in materials and method of revised manuscript.

Reviewer 2 Report

Main issues

The identification of comorbidities I understand that is done by a self-reported questionnaire fullfilled by participants (how they recorded comorbidities is not described in the Methods section) and this fact has an important bias (underestimating). 

There are some important factors as: smoke condition, drug abuse, obesity (IMC) or concomitant medication that are not recording and could be impacting on the findings. On the other hand, immunovirological situation of patients, in terms of pVL and CD4 nadir and current, as long as current ART, could be important to characterize HIV population and would be necessary to consider and include into the statistical analysis. 

Rate of female participants were significantly high (77.5%).  Maybe aspect related with sex (pregnancy, abortions, menopause status, etc) could have a role in the presence of comorbidities. Should be discussed.

It is remarkable and impressing the good findings in terms of QoL and adherence they have reported. But I would like to highlight that the Morisky medication adherence scale (MMAS-8) is reliable and valid measure to detect patients at risk of non-adherence, but is not exactly accurate to measure adherence. This result should by extended and explained. 

In the abstract, the phase of results “high level of comorbidities among PLWHA influenced by different factors as quality of life, general health and adherence to ART” is unclear, because “general health” is not described or considered as a variable in the study. 

Due to the HVB infection is the most prevalent comorbiditie, could be useful to extend some parameters about it, such as time of diagnosis, pVL, etc…

Minor issues. 

During discussion sections, mentions similar results in cohorts from Serbia and India. This cohorts could be really different form the Ghana cohort. Please, reconsider this point.

At the end of the discussion section, main author writes in first-person. Please be consistent with the rest of the manuscript language. 

MMAS-8 and BREF (WHOQOL-HIV) should be considered as Supplementary material. 

If RGDP was done during the data collection, should be mentioned in methods. 

The routine care description and follow-up frecuency in the clinic, could be useful to describe in the introduction section.

In my opinion, discussion section is too long, sometimes difficult to follow. 

Line 62 and 79

The authors already describe PLWHA, they can use the acronymous.

Line 96.

Statistical analysis plan should be better described. Same for p values. 

Line 117.

Please describe the meaning of symptomatic or asymptomatic for HIV.

Table 1.

Serostatus heading could be confusing to describe the presence of symtoms. Please re-consider to change it. 

Line 119.

Time living with HIV infection should be more accurate (since HIV infection? Since start ART?).

Line 137. 

Levels of adherence should be described in detailed. 

Line 323.

ARVs in not described previously.

Author Response

  1. A careful review of acronyms has been done to include relevant and appropriate acronyms throughout the text where necessary. 
  2. The statistical analysis plan has been duly stated and explained in the manuscript as recommended.
  3. Comorbidity has been defined under the materials and methods section. Information on how data on comorbidity was reported was documented as well as requested. Limitations recognized when compared to other works have been stated during discussion.
  4. Duration of years living with HIV has been clarified to be since infection.
  5. All factors could not be considered in this study but may be considered in future studies. Factors considered were based on literature reviews and how relatable they are to our setting. The closest to immunovirological situation of patients accounted for in our work was to see if participants reported symptoms of HIV or not. This was stated in our manuscript as a limitation during discussion.
  6. Rate of female participants being high is consistent with National data. However, since gender was not a major objective of the research work, data specific to pregnancy, abortions, menopause status, etc., were not collected thus, cannot be discussed. These can be considered in future studies.
  7. The MMAS-8 have been extended and explained in materials and methods thus, giving a clearer picture and explanation to results presented.
  8. General health is a component of quality of life thus was not described as a ‘stand alone variable’. However, it has been deleted from abstract for the avoidance of confusion.
  9. Parameters about Hep B infection such as time of diagnosis and viral load were not considered since it was not the focus of study. This may be considered in follow-up studies.
  10. The point about the comparison of our results to cohorts from Serbia and India was reconsidered as suggested and replaced with a Ghanaian cohort.
  11. Correction has been made to the end of the discussion, where first person was used, as pointed out.
  12. MMAS-8 and WHOQOL-HIV BREF has been considered and added to manuscript as suggested.
  13. The routine care description and follow-up frequency in the clinic has been described in the materials and methods section as recommended.
  14. Parts of the discussion sections have been deleted to minimize the length to make it easier for reading and understanding.
  15. Meaning of symptomatic and asymptomatic for HIV has been clearly described in materials and methods
  16. Serostatus heading in table 1 and subsequent tables has been changed to presence or absence of HIV symptoms to prevent confusion as suggested.
  17. Levels of adherence has been described in depth in materials and methods section thus bringing clarity to results.
  18. ART has been used in place of ARVs.

Round 2

Reviewer 1 Report

Reviewing the work was a pleasure, thank you for the invitation. 

This new handwritten version answered all the questions and added all my suggestions.

I believe it got published.

Congratulations

Reviewer 2 Report

Suggestions  have been incorporated correctly